# Universal entanglement signatures of foliated fracton phases

**Wilbur Shirley[1*], Kevin Slagle[2] and Xie Chen[1]**

**1** Department of Physics and Institute for Quantum Information and Matter,
California Institute of Technology, Pasadena, California 91125, USA
**2** Department of Physics, University of Toronto, Toronto, Ontario M5S 1A7, Canada

* wshirley@caltech.edu

## Abstract

Fracton models exhibit a variety of exotic properties and lie beyond the conventional framework of gapped topological order. In Ref. [1], we generalized the notion of gapped phase to one of *foliated fracton phase* by allowing the addition of layers of gapped two-dimensional resources in the adiabatic evolution between gapped three-dimensional models. Moreover, we showed that the X-cube model is a fixed point of one such phase. In this paper, according to this definition, we look for universal properties of such phases which remain invariant throughout the entire phase. We propose multipartite entanglement quantities, generalizing the proposal of topological entanglement entropy designed for conventional topological phases. We present arguments for the universality of these quantities and show that they attain non-zero constant value in non-trivial foliated fracton phases.

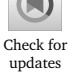
# 1 Introduction

Fracton models, a collection of gapped three-dimensional lattice models [2–15], are known to exhibit a range of exotic properties [16–25]. First, they harbor a ground state degeneracy (GSD) that is stable against arbitrary local perturbations and increases exponentially with linear system size. More strikingly, fracton models contain quasi-particle excitations whose motion is restricted to a sub-dimensional manifold (a plane or a line) or which cannot move individually at all. [2, 5, 8]. Due to these constraints on quasi-particle mobility, the models have unusually slow dynamics even in the absence of disorder [21, 22]. Furthermore, for the ground states of these models, the entanglement entropy of a region in the bulk contains a term that scales linearly with the size of the region, in addition to the dominant area law term which scales quadratically [23–25].

Among these properties, which ones are universal characteristics of fracton topological phases? This is an important question because the study of fracton phases thus far has been mostly focused on specific exactly solvable models. Once we move away from the exactly solvable points, we want to know which sets of properties remain and are indicative of the underlying fracton order. Moreover, given two generic interacting many-body models, we want to be able to determine whether or not they belong to the same fracton phase by comparing their universal properties.

For conventional (gapped) topological phases in 2D and 3D, such as fractional quantum Hall systems and discrete gauge theories, an understanding of the universal properties is more or less complete. These properties include the fractional quasi-particle content and their self- and mutual braiding statistics [26], the (finite) ground state degeneracy as a function of the topology of the spatial manifold [27, 28], the perimeter scaling law of Wilson loop operators in the ground state [29], the topological entanglement entropy [30–32] etc. At the same time, it is also clear that some properties of specific models are merely accidental and are not universal to the phase. Such accidental properties include—assuming there is no extra symmetry requirement on the models—a uniform Berry curvature in quantum Hall systems, the fact that electric and magnetic charges have the same energy in discrete gauge theories, an expectation value of unity for Wilson loops in the ground state, etc.

Fracton models lie beyond the conventional framework of gapped topological phases, which is made clear by the fact that their ground state degeneracy increases with system size. To extend the idea of universality to fracton models, we must first define the notion of a fracton phase. In Ref. [1], we generalized the notion of gapped topological phases to encompass fractons by allowing the addition of gapped two-dimensional resource layers when smoothly evolving between two three-dimensional gapped models. According to this definition, a stack of decoupled layers of 2D topological orders belongs to a trivial phase whereas the X-cube model belongs to a non-trivial phase [1]. It can be shown that the kagome lattice X-cube model [13], the checkerboard model [3], and the 3D toric code model (with trivial foliation structure) belong to non-trivial phases according to this definition as well. Due to the deep connection of this definition with the foliation structure of the underlying spatial manifold, we will refer to such phases as *foliated fracton phases*.

In accordance with this definition, in this paper we identify certain universal properties of these phases that remain invariant as one moves throughout each phase. We propose a multi-partite entanglement quantity (Fig. 3) calculated from the ground state wave function, generalizing the proposal of topological entanglement entropy [30–32] to characterize conventional topological orders. We argue for the universality of this quantity and show that it attains positive constant value (Table 1) in non-trivial phases that contain the X-cube model [3] on cubic and stacked-kagome lattices [13], the checkerboard model [3], the Chamon model [7], and the 3D toric code model [33] respectively. The multi-partite entanglement

quantity we design is in general non-topological in the sense that its value can change if the shape of the regions involved changes in an arbitrary way. However it does remain invariant provided it follows the foliation structure of the fracton model, which can be determined from simpler entanglement quantities calculated from the ground state wave function.

The paper is structured as follows. In Sec. 2, we review the definition of foliated fracton phases and explain its motivation and applicability. Based on this definition, in Sec. 3, we state the criteria that must be satisfied by an entanglement quantity in order to be universal. In Sec. 4, we present a scheme for calculating such a quantity and the calculation results for a handful of relevant models. We conclude with a discussion of open questions in section Sec. 5.

## 2   Foliated fracton phases

Foliated fracton phases are defined in Ref. [1] as follows: *Two gapped three dimensional Hamiltonians $H_1$ and $H_2$ are in the same **foliated fracton phase** if by adding layers of two-dimensional gapped Hamiltonians $H_1^{2D}$ to $H_1$, and layers of (potentially different) two-dimensional gapped Hamiltonians $H_2^{2D}$ to $H_2$, it is possible to adiabatically evolve from $H_1 + H_1^{2D}$ to $H_2 + H_2^{2D}$ without closing the gap.*[1]

Written as a formula, we have

$$\text{Foliated fracton phase:} \qquad H_1 + H_1^{2D} \xleftrightarrow{\text{Adiabatic evolution}} H_2 + H_2^{2D}. \qquad (1)$$

Here, adiabatic evolution refers to a smooth deformation of the Hamiltonian that preserves the energy gap, i.e. an evolution that does not pass through a critical point or an intervening gapless phase. Equivalently, because we are considering gapped systems, this relation can be stated in terms of the ground space. Denote by $GS_1$ and $GS_2$ the gapped ground spaces of $H_1$ and $H_2$, and $GS_1^{2D}$ and $GS_2^{2D}$ the gapped ground spaces of layers of 2D Hamiltonians. Then $H_1$ and $H_2$ are in the same foliated fracton phase if $GS_1 \otimes GS_1^{2D}$ and $GS_2 \otimes GS_2^{2D}$ can be mapped into each other through finite depth local unitary transformations. [1]

$$\text{Foliated fracton phase:} \qquad GS_1 \otimes GS_1^{2D} \xleftrightarrow[\text{unitary transformation}]{\text{Finite depth local}} GS_2 \otimes GS_2^{2D}. \qquad (2)$$

That is, although the finite depth local unitary acts on the entire Hilbert space, it must map ground states $GS_1 \otimes GS_1^{2D}$ into grounds states $GS_2 \otimes GS_2^{2D}$.

In comparison, the conventional definition of gapped phases only allows the addition of decoupled degrees of freedom in the form of a product state in the process of adiabatic evolution. That is, the conventional definition can be expressed as:

$$\text{Conventional gapped phase:} \quad H_1 + H_1^{0D} \xleftrightarrow{\text{Adiabatic evolution}} H_2 + H_2^{0D}, \qquad (3)$$

where $H_1^{0D}$ and $H_2^{0D}$ are Hamiltonians with direct product ground states. In terms of the ground space, the definition is given as

$$\text{Conventional gapped phase:} \quad GS_1 \otimes GS_1^{0D} \xleftrightarrow[\text{unitary transformation}]{\text{Finite depth local}} GS_2 \otimes GS_2^{0D}, \qquad (4)$$

where $G_1^{0D}$ and $G_2^{0D}$ are non-degenerate (one-dimensional as a Hilbert space) spaces spanned by respective product states. $GS_1$ and $GS_2$ are said to be connected by a 'generalized local unitary' (gLU) transformation [34].

---

[1]Before performing the adiabatic evolution (or local unitary transformations), we have the freedom to match locality and identify local degrees of freedom in the two models.

A major difference between these two definitions of phases of matter is that systems in the same conventional gapped phase always have the same GSD while systems in the same foliated fracton phase can have varying ground state degeneracy owing to the additional 2D layers. This simple observation is the chief motivation to propose this new definition as it is known that the GSD of fracton models can change with system size.

In Ref. [1], we showed that the X-cube model belongs to such a foliated fracton phase. The X-cube model is actually the fixed point of the phase that remains invariant under the renormalization group transformation: the X-cube model defined on a $L_x \times L_y \times L_z$ cubic lattice can be mapped to the X-cube model defined on a $L_x \times L_y \times (L_z + 1)$ cubic lattice by adding a layer of the 2D toric code in the $xy$ plane and applying local unitary transformations to sew this new layer into the original X-cube model. Similar procedures can be applied to increase the system size in the $x$ and $y$ directions as well. Therefore, the foliation structure of the X-cube model is composed of layers in the $xy$, $yz$ and $zx$ planes. Such a foliation structure provides a natural explanation for the linear scaling of the entanglement entropy and the logarithm of the GSD in the X-cube model. Similar RG transformations and foliation structures can be identified [35] in the kagome X-cube model [13] and the checkerboard model [3].

On the other hand, not all fracton models are captured by this notion of foliated fracton phase. Type-II fracton models such as the Haah code are evidently not encompassed by this definition as they do not contain two dimensional quasi-particles that can move freely in a plane. How to generalize these definitions to describe such fractal spin liquids remains an open question.

## 3 Signatures of long-range entanglement

Given the definition of foliated fracton phases, we can now pose the question of what universal properties characterize such phases and represent the corresponding foliated fracton order. In other words, we aim to identify properties of fracton models that remain invariant not only under smooth deformations of the Hamiltonian, but also under the addition or removal of gapped 2D layers.

As a first consideration, one can ask whether the ground state degeneracy (GSD) on a 3D torus plays such a role. In a conventional topological phase, the finite GSD (as a function of spatial topology) is indeed a universal quantity. Conversely, for foliated fracton phases, the GSD is no longer constant, but instead increases exponentially with linear system size and takes the generic form

$$\log \text{GSD} = aL + b. \tag{5}$$

This scaling form loses meaning in systems lacking a regular lattice structure (e.g. a general triangulation), for which it is not obvious how to measure $L$. Therefore, the GSD cannot serve as a universal quantity in the most general case. When translation symmetry is preserved, the constant $b$ is an invariant of the phase while $a$ does not have an absolute meaning as it can be arbitrarily changed by changing the unit of length. In the presence of translation invariance, $b$ can potentially be used to distinguish between different foliated fracton phases, although it only applies when the system exists on a three-torus and depends sensitively on the periodic boundary conditions.

We note that one aspect is in need of clarification: in Ref. [1], we discussed the scaling of the GSD (in the form of Eq. 5) of the X-cube model on various spatial 3-manifolds, and how its dependence can be interpreted as a consequence of the topology of the foliating leaves and the topology of their intersections. This discussion applies to the fixed point models studied in Ref. [1]. Away from the fixed point, however, both constants $a$ and $b$ may lose their meaning: $a$ becomes ill-defined due to the arbitrariness in choosing the unit of length, whereas $b$ is not

well-defined due to the existence of 'small' logical operators near singularities which do not have an infinite size in an infinite system.[2]

Alternatively, we aim to identify universal quantities that do not depend on boundary conditions or translation invariance. For conventional topological phases, the topological entanglement entropy [30–32] is known to be such a quantity, and can be calculated from a local region in a ground state wave function. In this paper, we seek to characterize foliated fracton phases using a similar quantity. In the following subsections, we first briefly review the notion of topological entanglement entropy and then specify explicit criteria that must be satisfied by an entanglement quantity in order to universally characterize foliated fracton order. In section 4, we present such a quantity.

### 3.1 Review of topological entanglement entropy

Recall that the entanglement entropy of a state $|\psi\rangle$ with respect to a region $R$ is defined as the von Neumann entropy

$$S_R = -\mathrm{tr}(\rho_R \log \rho_R) \tag{6}$$

of the reduced density operator $\rho_R = \mathrm{tr}_{\overline{R}} |\psi\rangle \langle \psi|$ where the subsystem $\overline{R}$, the complement of $R$, has been traced out. Because the model Hamiltonians we discuss are $Z_2$ stabilizer codes, it is convenient to take logarithms with respect to base 2 throughout the paper. For ground states of gapped 2D systems, the entanglement entropy takes the generic form

$$S_R = \alpha L - c\gamma + \dots , \tag{7}$$

where $L$ is the length of the boundary $\partial R$, $c$ is the number of connected components of $\partial R$, $\alpha$ is a non-universal constant, and the region $R$ is assumed to have a smooth boundary relative to the correlation length of the system [36, 37]. (The ellipsis represents contributions that vanish when $L$ is large and $\partial R$ is smooth.) Whereas the dominant area law term is sensitive to the microscopic details of the model, the topological contribution $-c\gamma$ is a universal feature of generic topologically ordered ground states, and is referred to as the topological entanglement entropy. Here $\gamma = \log \mathcal{D}$ where $\mathcal{D}$ is the total quantum dimension of the 2D topological order [30, 31]. For non-chiral orders, the origin of this term has a simple interpretation in terms of the string-net condensate picture [38] of 2D topological ground state wavefunctions: the net topological charge of all strings crossing a component of $\partial R$ must be trivial, resulting in non-local correlations that correspondingly reduce the entropy of entanglement [31].

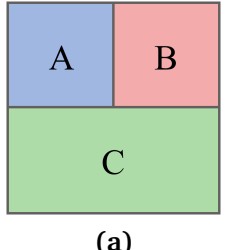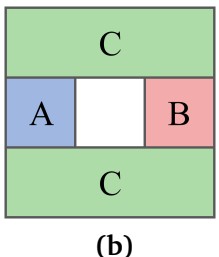

**(a)**             **(b)**

Figure 1: **(a)** Square $I(A;B;C)$ and **(b)** annular $I(A;B|C)$ schemes to isolate topological entanglement entropy in 2D.

It is possible to isolate the topological entanglement entropy $-c\gamma$ by taking additive combinations of entanglement entropies of varying regions suitably chosen to cancel the area law terms as well as local contributions that may arise from sharp corners in the boundary of a region [30, 31]. Two such schemes for extracting the topological term are depicted in Fig. 1.

---

[2]These 'small' logical operators occur in, for example, the foliation of $S^2 \times S^1$ considered in Ref. [1].

In each, three compact regions (*A*, *B*, and *C*) with partially shared boundary are carved out of the planar medium. For the square scheme (Fig. 1(a)), the quantum tripartite information

$$I(A;B;C) \equiv S_A + S_B + S_C - S_{AB} - S_{BC} - S_{AC} + S_{ABC} \qquad (8)$$

is used. *AB* denotes the composite of regions *A* and *B*. Each region's entropy contributes a single $-\gamma$ term, so in total $I(A;B;C) = -\gamma$ [30]. In the annular scheme (Fig. 1(b)), the tripartite information reduces to the simpler expression for the quantum conditional mutual information

$$I(A;B|C) \equiv S_{AC} + S_{BC} - S_C - S_{ABC}. \qquad (9)$$

Since the boundaries of regions *C* and *ABC* each have two components, it follows that $I(A;B|C) = 2\gamma$ [30,31]. Crucially, these entanglement quantities remain unchanged under generalized local unitary (gLU) transformations (Eq. 4) of the ground state. In this sense, they represent universal signatures of the long-range entanglement structure of 2D topological orders, and can be used to detect the order present in generic ground state wavefunctions away from the RG fixed-point. Moreover, these quantities are topological invariants; i.e. they depend solely on the connectivity of the regions and not on their geometry.

For ground states of gapped phases in 3D, the entanglement entropy of a region *R* takes the generic form

$$S_R = \alpha A + \beta L + \gamma + \dots , \qquad (10)$$

where in addition to the area law term $\alpha A$ ($\alpha$ is a non-universal constant and *A* is the area of the boundary $\partial R$), a subleading correction, $\beta L$, linear in the length *L* of the region may be present [39,40]. The constant term $\gamma$ contains both universal corrections as well as non-universal local contributions due to the curvature of $\partial R$ (manifesting in a correction proportional to $\chi$, the Euler characteristic of $\partial R$) [32,41]. For conventional gapped topological phases in 3D, the linear corrections vanish, and suitable generalizations of the 2D *ABC* schemes serve as entanglement signatures of the topological order [32,33].

## 3.2 Subleading linear corrections and foliation structure

Conversely, for foliated fracton phases (as well as simple decoupled stacks of 2D topological orders) the subleading linear corrections can not be ignored. Previous work has employed similar schemes (Fig. 2) to isolate these linear contributions from the dominant area law term [23,24].

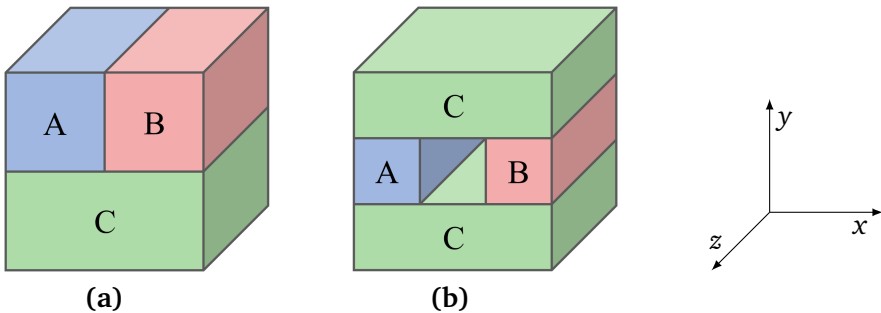

**(a)**            **(b)**

Figure 2: **(a)** 3D cube $I(A;B;C)$ and **(b)** solid torus $I(A;B|C)$ schemes. In both cases the regions are contained within an overall cube of side length *L*.

The results of these prescriptions are elucidated by the underlying foliation structure of the fracton models; the surviving linear quantity is an additive combination of topological entanglement entropies from the individual foliating layers. For example, applying the cube

and solid torus schemes (Fig. 2) to the X-cube model yields the quantities $I(A;B;C) = -L$ and $I(A;B|C) = 2L + 1$, respectively [24]. The linear components arise from the toric code foliating layers parallel to the $xy$ plane, which intersect the cube and solid torus schemes in the respective 2D square and annular schemes (Fig. 1), and thus contribute $-1$ and 2 per unit length to these quantities respectively (as the total quantum dimension of the toric code topological order is $\mathcal{D} = 2$). The foliation perspective of fracton phases therefore suggests that the linear term in entanglement entropy is itself a non-universal feature of specific models, as it absorbs the topological entanglement entropies of added layers of 2D topological orders. Thus, we argue that these sub-extensive entanglement quantities are not universal. (The constant component, excluding the curvature contribution, is not universal either, which was pointed out in [24].)

These schemes can, however, be used to diagnose the underlying foliation structure. Consider a model with underlying foliations labelled $i = 1, 2, \ldots, n$, where foliation $i$ is composed of parallel leaves with separation $1/|\mathbf{F}_i|$ and orthogonal to the vector $\mathbf{F}_i$. Each leaf is composed of a 2D topological order with topological entanglement entropy $\gamma_i = \log \mathcal{D}_i$. Then consider a tripartite cube scheme (Fig. 2(a)) described by a vector $\mathbf{L}$. For this scheme the overall cube has side length $|\mathbf{L}|$, and the front face is normal to $\mathbf{L}$. Then

$$I(A;B;C) = -\sum_i \gamma_i |\mathbf{L} \cdot \mathbf{F}_i| + O(1). \tag{11}$$

Due to the non-linearity of this expression, the orientations of the underlying foliations can be deduced by considering several such tripartite cubic schemes with varying overall orientation.

For instance, consider the X-cube model. As discussed, a cubic scheme of size $L$ with the front face oriented normal to the $x$, $y$, or $z$ direction will result in a tripartite information of $I(A;B;C) = -L$. However, rotating the regions such that the front face of the cube is normal to the $(1,1,1)$ direction yields $I(A;B;C) \sim -L\sqrt{3}$. These results are consistent with a foliation structure aligned parallel to the $xy$, $yz$, and $xz$ planes. (In order to rule out all other possible foliation structures, schemes with additional orientations would have to be examined in order to check consistency.) Conversely, for the X-cube model on the stacked kagome lattice, there are four underlying foliations. The stacked kagome lattice is built out of a stacked triangular Bravais lattice with basis vectors $\hat{x}, \hat{z}$, and $\alpha = (1/2, \sqrt{3}/2, 0)$. A cube scheme with the front face normal to the $z$ direction yields $I(A;B;C) = -L$, whereas schemes with the front face parallel to the planes spanned by $\hat{z}$ and $\hat{x}$, $\hat{z}$ and $\alpha$, or $\hat{z}$ and $\alpha - \hat{x}$ will each yield $I(A;B;C) \sim -2L/\sqrt{3}$.

### 3.3 Criteria for universal entanglement quantity

In pursuit of universal characteristics, we are motivated to take an additional step and identify an entanglement quantity $I$ that satisfies the following criteria:

1. All area law and local contributions to $I$ cancel.

2. All contributions to $I$ from the foliating layers must cancel. (This would otherwise result in contributions that scale linearly with subsystem size.)

3. $I$ attains non-zero value for non-trivial foliated fracton phases (including conventional topological phases).

We note that the first and second criteria together are equivalent to demanding that $I$ vanishes for arbitrary product states as well as simple decoupled stacks of 2D topological states. Thus, in accordance with the definition of foliated fracton phases discussed in the previous section, these criteria merely codify the requirement of a universal quantity that it is invariant under gLU transformations augmented with the free addition or removal of 2D topological resource states.

# 4  Universal schemes for foliated fracton phases

In this section we introduce a family of novel entanglement schemes with the above criteria in mind, and apply these prescriptions to a handful of stabilizer code models, revealing universal signatures of foliated fracton order.

## 4.1  Wireframe schemes

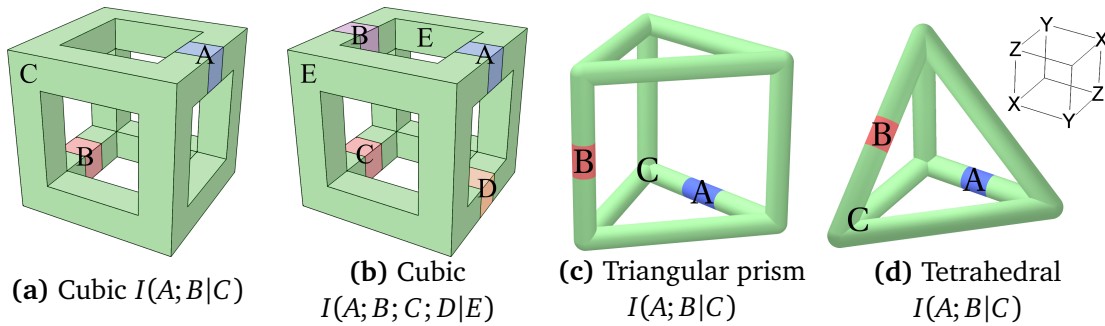

**(a)** Cubic $I(A;B|C)$  **(b)** Cubic $I(A;B;C;D|E)$  **(c)** Triangular prism $I(A;B|C)$  **(d)** Tetrahedral $I(A;B|C)$

Figure 3: **(a)** Cubic $I(A;B|C)$, **(b)** cubic $I(A;B;C;D|E)$, **(c)** triangular prism $I(A;B|C)$, and **(d)** tetrahedral $I(A;B|C)$ entanglement schemes for foliated fracton phases. **(d)** Stabilizer for the Chamon model defined on a cubic lattice with one qubit per vertex (inset).

These prescriptions employ a set of regions whose union forms a solid wireframe region which is aligned with the foliating layers and supports closed branching string operators in the shape of the wireframe. The quantities considered are the quantum conditional mutual information $I(A;B|C)$ and the quantum conditional four-partite information $I(A;B;C;D|E)$ for choices of regions depicted in Fig. 3. By definition, $I(A;B;C;D|E) = I(A;B;C;D) - I(A;B;C;D;E)$ where $I(A;B;C;D)$ and $I(A;B;C;D;E)$ are the quantum four-partite and five-partite information respectively. Explicitly,

$$
\begin{aligned}
I(A;B;C;D|E) \equiv &-S_E + S_{AE} + S_{BE} + S_{CE} + S_{DE} \\
&- S_{ABE} - S_{BCE} - S_{CDE} - S_{ACE} - S_{BDE} - S_{ADE} \\
&+ S_{ABCE} + S_{ABDE} + S_{ACDE} + S_{BCDE} - S_{ABCDE}.
\end{aligned}
\tag{12}
$$

$I(A;B|C)$ is defined in (9). Following the arguments of Refs. [30, 31], these schemes directly cancel the area law and local contributions of each boundary region. As discussed in the previous section, to ensure the cancellation of the subleading linear corrections, it is sufficient to guarantee that no foliating layer contributes a non-zero topological entropy. Each of our schemes is designed such that no layer intersects all regions of the scheme, ensuring that the contributions of each foliating layer, to the quantities $I(A;B|C)$ and $I(A;B;C;D|E)$, vanish. These quantities thus capture a universal feature of the long-range entanglement structure.

We have computed their values numerically for the stabilizer code models listed using the methods introduced in Ref. [42]. (Details of these computations are contained in Appendix A; a review of the models considered is contained in Appendix B.) The results are summarized in Table 1. As can be seen, the tetrahedral and triangular-prism schemes yield non-zero values only for the Chamon model and the stacked kagome X-cube model respectively, owing to their unique foliation structures.

For the 3D toric code, these values can be understood in terms of the string condensate picture of the ground state wavefunction. Given a region $R$, each component of $\partial R$ must be pierced by an even number of strings, which decreases the Schmidt number

Table 1: Entanglement quantities for the wireframe schemes discussed (Fig. 3). Logarithms (in Eq. (6)) are calculated in base 2. Models are reviewed in Appendix B. [†]In order to attain a non-zero value for the kagome lattice X-cube model [13], the regions must be slanted in accordance with the foliation structures so that the wireframe actually forms a parallelepiped (see Fig. 4(c)). [‡]Here we have modified the Chamon model so that it is defined on a cubic lattice with one qubit per vertex. The lone stabilizer term is depicted in Fig. 3(d).

| | Cubic $I(A;B\vert C)$ | Cubic $I(A;B;C;D\vert E)$ | Triangular prism $I(A;B\vert C)$ | Tetrahedral $I(A;B\vert C)$ |
|---|---|---|---|---|
| 2D toric code stack | 0 | 0 | 0 | 0 |
| 3D toric code | 0 | 1 | 0 | 0 |
| X-cube model | 1 | 1 | 0 | 0 |
| Kagome X-cube[†] | 1 | 1 | 1 | 0 |
| Checkerboard model | 2 | 2 | 0 | 0 |
| Chamon model[‡] | 1 | 1 | 0 | 1 |

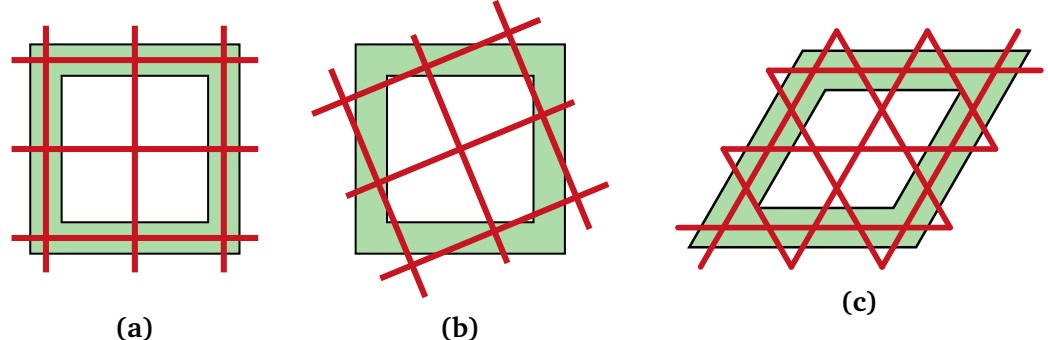

**(a)**  **(b)**  **(c)**

Figure 4: A side-view of the cubic entanglement regions (green) from Fig. 3(a-b) for different possible orientations with respect to the foliating layers (red). **(a)** Proper alignment on a cubic lattice, yielding the values Table 1. **(b)** Improper alignment, for which entanglement quantities $I(A;B\vert C) = I(A;B;C;D\vert E) = 0$. **(c)** Top-down view of a properly aligned solid wireframe on a stacked-kagome lattice, which yields $I(A;B\vert C) = I(A;B;C;D\vert E) = 1$ for the kagome X-cube model [13] as per Table 1.

of the reduced density operator $\rho_R$ by 1 and thus contributes $-1$ to $S_R$ per boundary component. The three-region wireframe schemes each contain four positive and four negative topological contributions, and hence $I(A;B\vert C)$ vanishes, whereas in the five-region scheme $I(A;B;C;D\vert E) = 1$ due to nine positive contributions and eight negative contributions.

Intriguingly, for the foliated fracton models, this relation no longer holds, implying that the universal contribution to the entanglement entropy of a region is not simply proportional to the number of boundary components. Moreover, we find that $I(A;B\vert C)$ and $I(A;B;C;D\vert E)$ are not invariants of the region topology, but rather depend intimately on their geometry; for example, simply rotating the overall figures such that the wireframes do not align with the axes of the foliation structure causes both quantities to vanish for all of the foliated fracton models considered (see Fig. 4). However, the quantities are invariant under changes in the overall size and thickness of the wireframe as well as generic 'small' deformations of the regions.

## 4.2 Lower bounds on conditional mutual information

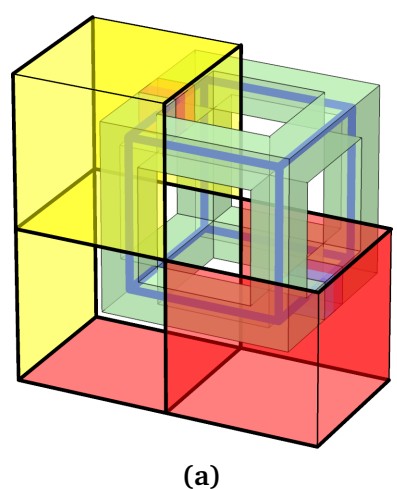
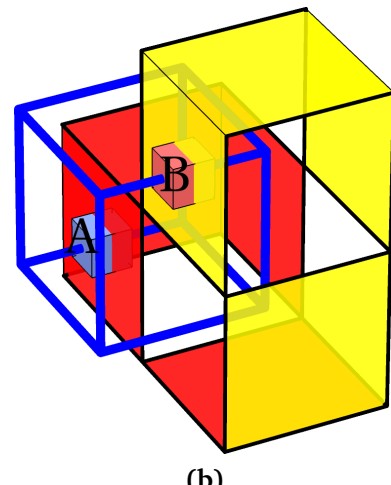

**(a)**                           **(b)**

Figure 5: Operators satisfying the conditions of Eq. (13) which can be used to bound $I(A;B|C)$ for the cubic scheme depicted in Fig. 3(a). **(a)** For the X-cube model (Fig. 7(c)), $I(A;B|C) \geq 1$ is obtained by taking $W_1$ to be a product of $X$ operators along the blue lines, and $U_1$ and $U_1^{\text{def}}$ to be products of $Z$ operators over all links that penetrate the red and yellow regions, respectively. For the checkerboard model (Fig. 7(d)), $I(A;B|C) \geq 2$ can be obtained by taking $W_1$ ($W_2$) to be a product of $X$ ($Z$) operators along the blue lines, and $U_1$ and $U_1^{\text{def}}$ ($U_2$ and $U_2^{\text{def}}$) to be products of $Z$ ($X$) operators over the red and yellow surfaces, respectively. $I(A;B|C) \geq 1$ can similarly be obtained for the Chamon model (Fig. 7(e)) using a tetrahedal-shaped geometry, but each operator will contain a mix of $X$, $Y$, and $Z$ Pauli operators. **(b)** Another view with subsystem $C$ (green) hidden for clarity.

The existence of closed branching string operators supported by the solid wireframe shape can be used to establish a lower bound on the conditional mutual information $I(A;B|C)$ via the methods introduced in Ref. [23]. These bounds are saturated by the values reported in Table 1. In particular, given the existence of unitary operators $U_i$, $U_i^{\text{def}}$, and $W_i$ (for $i = 1, \ldots, n$) that satisfy the following conditions:

$$
\begin{aligned}
&U_i \subset \overline{AC} && U_i |\psi\rangle = U_i^{\text{def}} |\psi\rangle && U_i W_i = -W_i U_i \\
&U_i^{\text{def}} \subset \overline{BC} && W_i |\psi\rangle = |\psi\rangle && U_i W_j = W_j U_i \;\; \text{if} \;\; i \neq j \\
&W_i \subset ABC,
\end{aligned}
\tag{13}
$$

where $|\psi\rangle$ is a ground state of the model and $U_i \subset \overline{AC}$ indicates that $U_i$ has support in $\overline{AC}$, the following inequality is satisfied [23]:

$$
I(A;B|C) \geq n.
\tag{14}
$$

For the fracton models considered under our schemes, $W_i$ can be chosen to be a closed branching string operator in the shape of the wireframe, piercing open membrane operators $U_i$ and $U_i^{\text{def}}$, which create fractonic excitations in identical locations. As an example, in Fig. 5 we depict unitary operators that apply to the cubic entanglement scheme for the X-cube, checkerboard, and Chamon models.

## 5 Discussion

In this paper, we have identified multi-partite entanglement quantities that represent universal signatures of zero-temperature foliated fracton order, and thus characterize the corresponding foliated fracton phases. These schemes are borne of the observation that layers of 2D topological orders serve as resources in the RG transformations for certain fracton models. These layers constitute an underlying foliation structure which is, by design, invisible to the entanglement quantities we consider. The non-zero values they attain for the X-cube, kagome lattice X-cube, and checkerboard models are a manifestation of the non-trivial long-range entanglement structure present in the ground states of these exotic phases of matter. Nonetheless, an understanding of the universal properties of these phases is still far from complete. Whereas for conventional topological orders a complete picture of universal characteristics is described in terms of quasiparticle sectors and their braiding statistics, elegantly packaged in the framework of topological quantum field theory (TQFT), it remains unclear which set of properties fully characterize foliated fracton orders, and what mathematical framework underlies the classification of these phases.

On the other hand, the related fractal spin liquids, i.e. type-II fracton models, remain largely enigmatic. To begin with, it is not clear in what sense these models even represent phases of matter. What is apparent is that, like conventional topological orders and foliated fracton orders, the ground states of fractal spin liquids exhibit highly non-trivial patterns of long-range entanglement. It remains an open question whether the entanglement structure present in these models can be captured by similar universal quantities. The foliated fracton models are also related (via Higgs and partial confinement mechanisms [43, 44]) to higher-rank $U(1)$ gauge theories with fractonic charge excitations [45–53]. The entanglement structure of these gapless models [54] is another potentially interesting avenue of future research.

## Acknowledgements

**Funding information** W.S. and X.C. are supported by the National Science Foundation under award number DMR-1654340 and the Institute for Quantum Information and Matter at Caltech. X.C. is also supported by the Alfred P. Sloan research fellowship and the Walter Burke Institute for Theoretical Physics at Caltech. K.S. is supported by the NSERC of Canada and the Center for Quantum Materials at the University of Toronto.

## A Numerical calculations

In this appendix, we briefly discuss details of the numerical calculations used to obtain the results of Table 1. Each model considered is a stabilizer code model, i.e. the Hamiltonians are sums of products of Pauli matrices where each term commutes with all other terms and has eigenvalue $-1$ in the ground state. For this class of exactly solvable models, entanglement entropies can be computed numerically (in polynomial time with subsystem size) using the methods introduced in Ref. [42]. Fig. 6 illustrates the lattice geometries that realize the entanglement subsystems in Fig. 3. For the toric code and X-cube models, multiple edges are grouped together appropriately to form unit cells.

As discussed in the main text, it is crucial that the subsystems are aligned with the foliating layers (see Fig. 4). For the X-cube and checkerboard models, the foliating layers are the $xy$, $xz$, and $yz$ planes; therefore the cubic entanglement schemes must be aligned with these axes

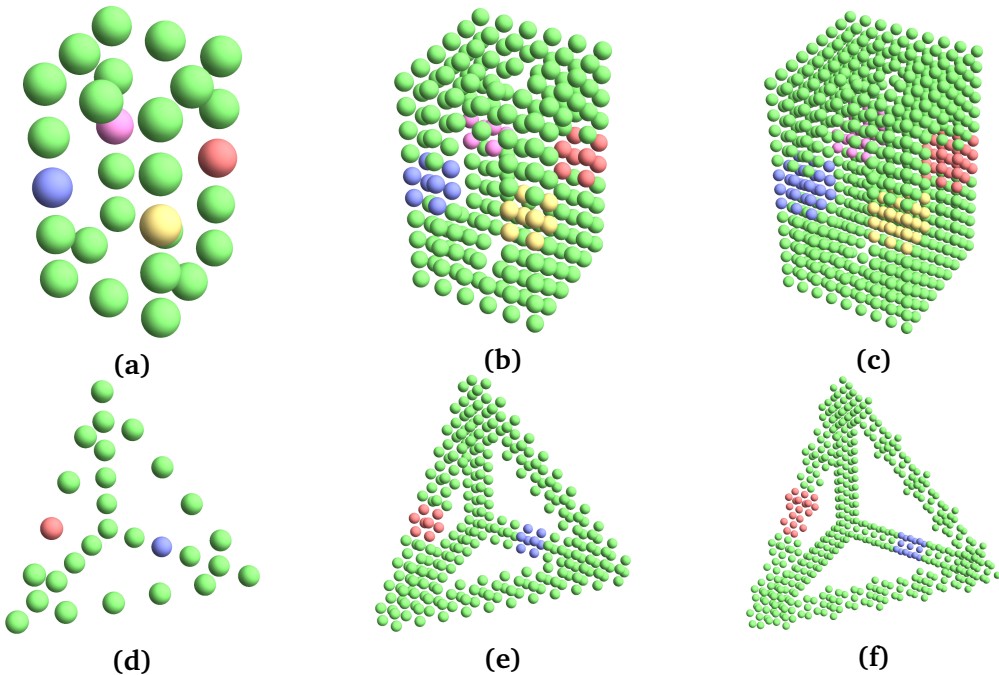

Figure 6: Geometries used to check Table 1 on a computer. Each point represents a unit cell of the lattice model that is included in the subsystem of the given color. **(a-c)** Lattice implementations of the cubic $I(A;B;C;D|E)$ scheme for three different sizes of subregions. The cubic and triangular-prism $I(A;B|C)$ scheme implementations are similar. **(d-f)** Lattice implementations of the tetrahedral $I(A;B|C)$ scheme. Since we used stabilizer models, no entry in Table 1 depended on the subsystem sizes.

(as per Fig. 4(a-b)). For the X-cube model on the stacked kagome lattice, the cubic wireframe must be tilted so that it is actually a parallelepiped in order to yield a non-zero value (see Fig. 4(c)), and similarly the triangular prism must be aligned with the underlying stacked triangular Bravais lattice. For the Chamon model, it is actually convenient to redefine the model on a cubic lattice (as in Fig. 3(d)). For liquid topological models, such as the 3D toric code, there is no (non-trivial) foliation structure and thus the orientation of the subsystems does not matter.

# B   Model Hamiltonians

This appendix contains a review of the models discussed in the paper. Each of these models is a qubit stabilizer code, meaning that the Hamiltonian is composed of mutually commuting products of Pauli operators.

The 2D toric code, originally introduced in [55], is defined on a square lattice with one qubit per edge, and has Hamiltonian

$$H = -\sum_{v} A_v - \sum_{p} B_p \,, \tag{15}$$

where $v$ runs over all vertices and $p$ runs over all elementary plaquettes, $A_v$ is a product of Pauli $Z$ operators over the edges adjacent to $v$, and $B_p$ is a product of Pauli $X$ operators over the edges of $p$ (see Fig. 7(a)). As a natural generalization, the 3D toric code has one qubit degree of freedom on each edge of a cubic lattice. The Hamiltonian is defined similarly, except

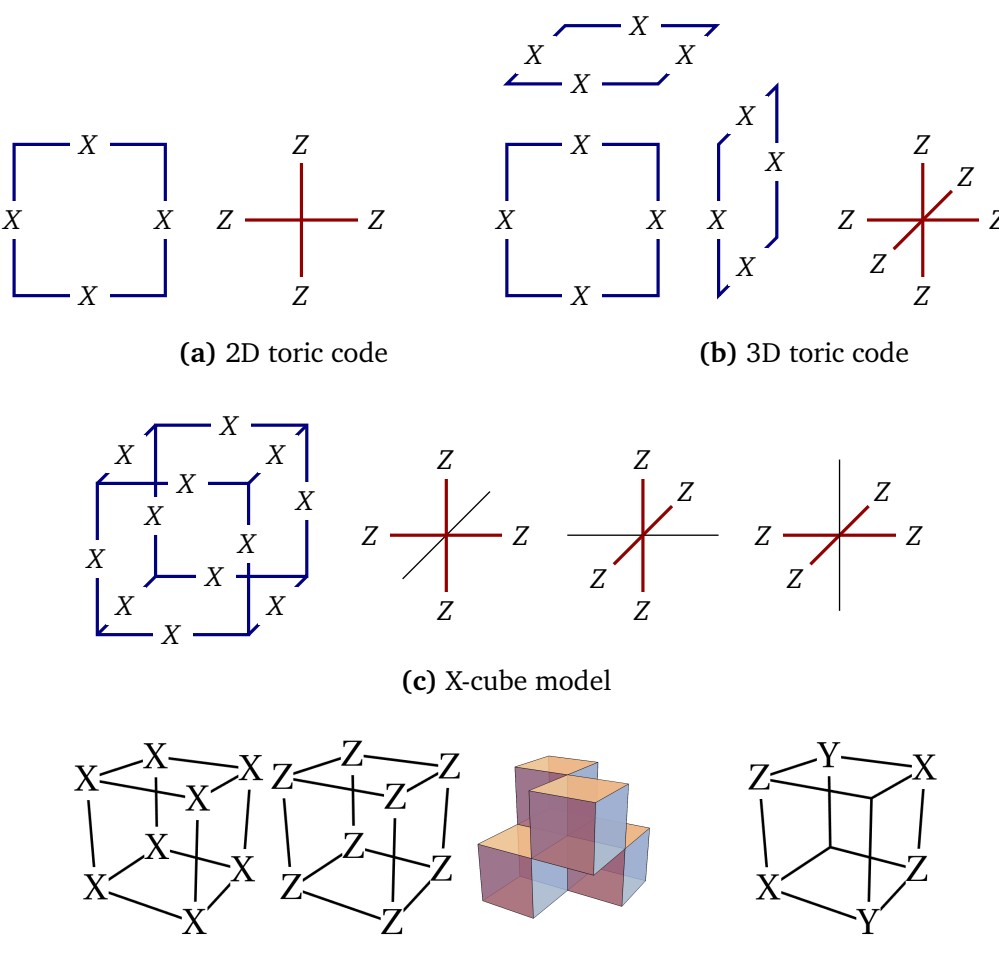

**(a)** 2D toric code  **(b)** 3D toric code

**(c)** X-cube model

**(d)** Checkerboard  **(e)** Chamon model

Figure 7: Hamiltonian terms for the stabilizer code models discussed.

there are three orientations of plaquettes, and the $A_v$ term is modified to a six-spin interaction (as shown in Fig. 7(b)).

The X-cube model is likewise defined on a cubic lattice with one qubit per edge. The Hamiltonian takes the form

$$H = -\sum_v \left( A_v^{xy} + A_v^{yz} + A_v^{zx} \right) - \sum_c B_c \,, \tag{16}$$

where $v$ runs over vertices and $c$ runs over elementary cubes. Here $A_v^{xy}$ is a product of $Z$ operators over the edges adjacent to $v$ in the $x$ and $y$ directions, whereas $B_c$ is a product of $X$ operators over all edges of $c$ (see Fig. 7(c)). As discussed in [13], it is possible to generalize the X-cube model to a stacked kagome lattice (as well as other lattice geometries), again with one qubit per edge. As each vertex of the stacked kagome lattice is locally isomorphic to a cubic lattice vertex, in the generalized model there remain three vertex terms which are fourfold products of $Z$ operators. However, the cube terms are replaced by generalized 3-cell terms for each elementary volume of the lattice (triangular and hexagonal prisms), which are likewise products of $X$ operators over all edges of the 3-cell.

Finally, the checkerboard and Chamon models are both defined on a cubic lattice with one qubit per vertex. For the checkerboard model, the elementary cubes are divided into 3D checkerboard *A-B* sublattices. The Hamiltonian is composed of two terms for each cube in the *A* sublattice: the first is a product of Pauli $X$ operators over all vertices of the cube, whereas

the second is a product of Pauli $Z$ operators over all vertices of the cube (see Fig. 7(d)). For the Chamon model, there is one stabilizer term per unit cell, as depicted in Fig. 7(e).

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
