# Peer review of "Universal entanglement signatures of foliated fracton phases"

_SciPost Physics, doi:SciPost Phys. 6, 015 (2019)_

## Round 1 · Referee Report · Anonymous (Referee 1) · 2018-6-21

Strengths

1. Clear calculations
2. Well written

Weaknesses

1. Narrow focus
2. Interest to non-experts unclear.

Report

This is a solid and well written paper on an interesting topic. I recommend publication as is.

Requested changes

None

---

## Round 1 · Referee Report · Anonymous (Referee 2) · 2018-6-23

Strengths

1) this paper addresses the open question of understanding entanglement in foliated fracton phases

2) the authors study the different contribution to the entanglement entropy and propose schemes to distil a universal contribution, using multipartite entanglement measures

3) they show that the proposed universal signatures are constant throughout foliated fracton phases, by explicit calculation in a handful of model Hamiltonians

Weaknesses

1) it is not clear what to make of these universal signatures, why are they important, and whether they will have an impact on the related area of research. Further work is perhaps needed to better understand their significance, and possibly their relation to the structure of the excitations

Report

the paper is well structured and well written, accessible to a reader with some background on topological lattice models, entanglement entropy, and fracton phases. The results are valid, to the best of my understanding, and deserve publication in SciPost.

Requested changes

1) to improve the accessibility of the paper to a broader audience, the authors could perhaps spend a few words to explain what they mean by the double arrow "adiabatic evolution" in Eq.(1)

2) proofreading for typos: "the the" and "fracon". Missing period at the end of Eq.(5).

3) I think that the von Neumann entropy in Eq.(6) has the wrong sign

4) at the end of Sec.3.1, the authors cite Ref.32 on two occasions. I wonder if earlier references may be more appropriate here (at least in addition to Ref.32). For example, in relation to non-universal contributions due to the Euler characteristic, PRL 97, 050404 (2006); and for entanglement signatures of gapped 3D topological phases, Ref.33

---

## Round 3 · Author Response

We thank the referees for their reports and Referee #2 for their constructive comments.

---

## Round 3 · List of Changes

We have made the following changes in accordance with Referee #2's suggestions.

1) to improve the accessibility of the paper to a broader audience, the authors could perhaps spend a few words to explain what they mean by the double arrow "adiabatic evolution" in Eq.(1)

We have added the following sentence to explain what is meant by adiabatic evolution: "Here, adiabatic evolution refers to a smooth deformation of the Hamiltonian that preserves the energy gap, i.e. one that does not pass through an intervening gapless phase."

2) proofreading for typos: "the the" and "fracon". Missing period at the end of Eq.(5).

These typos have been fixed.

3) I think that the von Neumann entropy in Eq.(6) has the wrong sign

This error has been corrected.

4) at the end of Sec.3.1, the authors cite Ref.32 on two occasions. I wonder if earlier references may be more appropriate here (at least in addition to Ref.32). For example, in relation to non-universal contributions due to the Euler characteristic, PRL 97, 050404 (2006); and for entanglement signatures of gapped 3D topological phases, Ref.33

We have added citation of the suggested references on both occasions.

---

## Editorial Decision

published